# Molecular Biomarkers in Neurological Diseases: Advances in Diagnosis and Prognosis

**DOI:** 10.3390/ijms26052231

**Published:** 2025-03-01

**Authors:** Athena Myrou, Konstantinos Barmpagiannos, Aliki Ioakimidou, Christos Savopoulos

**Affiliations:** 1Department of Internal Medicine, American Hellenic Educational Progressive Association (AHEPA) University Hospital, 54636 Thessaloniki, Greece; mparmpak@auth.gr (K.B.);; 2Microbiology Laboratory, Department of Immunology, American Hellenic Educational Progressive Association (AHEPA) University Hospital, 54636 Thessaloniki, Greece; aliki_ioakimidou@yahoo.gr

**Keywords:** molecular biomarkers, neurological diseases, multi-omics, liquid biopsy, precision medicine, neurodegeneration, neuroinflammation

## Abstract

Neurological diseases contribute significantly to disability and mortality, necessitating improved diagnostic and prognostic tools. Advances in molecular biomarkers at genomic, transcriptomic, epigenomic, and proteomic levels have facilitated early disease detection. Notably, neurofilament light chain (NfL) serves as a key biomarker of neurodegeneration, while liquid biopsy techniques enable non-invasive monitoring through exosomal tau, α-synuclein, and inflammatory markers. Artificial intelligence (AI) and multi-omics integration further enhance biomarker discovery, promoting precision medicine. A comprehensive literature review was conducted using PubMed, Scopus, and Web of Science to identify studies (2010–2024) on molecular biomarkers in neurodegenerative and neuroinflammatory disorders. Key findings on genomic mutations, transcriptomic signatures, epigenetic modifications, and protein-based biomarkers were analyzed. The findings highlight the potential of liquid biopsy and multi-omics approaches in improving diagnostic accuracy and therapeutic stratification. Genomic, transcriptomic, and proteomic markers demonstrate utility in early detection and disease monitoring. AI-driven analysis enhances biomarker discovery and clinical application. Despite advancements, challenges remain in biomarker validation, standardization, and clinical implementation. Large-scale longitudinal studies are essential to ensure reliability. AI-powered multi-omics analysis may accelerate biomarker application, ultimately improving patient outcomes in neurological diseases.

## 1. Introduction

Neurological diseases, including Alzheimer’s disease (AD), Parkinson’s disease (PD), amyotrophic lateral sclerosis (ALS), and multiple sclerosis (MS), pose substantial health and economic challenges worldwide. The increasing prevalence of these disorders, combined with their complex pathophysiology, necessitates the development of reliable molecular diagnostics [1]. Traditional diagnostic methods often rely on clinical symptoms and imaging findings, which may not be sufficient for early disease detection or prognosis [2]. Consequently, molecular biomarkers have emerged as powerful tools that provide insights into disease mechanisms, enhance diagnostic accuracy, and guide therapeutic interventions [3].

Advancements in omics technologies, including genomics, transcriptomics, epigenomics, proteomics, and metabolomics, have led to the identification of novel biomarkers for neurological diseases [4]. Genomic studies have uncovered key genetic risk factors and mutations associated with disease susceptibility, while transcriptomic analyses have highlighted dysregulated gene expression patterns [5]. Epigenetic modifications, such as DNA methylation and histone modifications, provide insights into disease progression and gene regulation [6]. Furthermore, proteomic and metabolomic approaches have refined our understanding of neurodegenerative and neuroinflammatory processes, identifying disease-specific proteins and metabolic alterations that may serve as potential biomarkers for early detection and prognosis [7].

Recent technological developments, such as liquid biopsy and multi-omics data integration, have facilitated non-invasive biomarker detection and improved disease classification [8]. Liquid biopsy techniques allow for the analysis of circulating nucleic acids, proteins, and extracellular vesicles, providing a minimally invasive approach to disease monitoring [9]. Multi-omics integration, combining data from different molecular levels, has uncovered complex interactions underlying neurological conditions, paving the way for precision medicine strategies [10].

This review explores the latest progress in molecular biomarker discovery, emphasizing its clinical applications in diagnosing and monitoring neurological diseases. By addressing the current challenges and future directions in the field, we aim to highlight the translational potential of molecular diagnostics in improving patient outcomes [11].

## 2. Results

### 2.1. Genomic and Transcriptomic Biomarkers

Genetic mutations, single nucleotide polymorphisms (SNPs), and transcriptomic alterations have been implicated in various neurological diseases, emphasizing the role of genetic predisposition and gene expression dysregulation in neurodegeneration and neuroinflammation. Advances in next-generation sequencing (NGS) technologies have significantly improved our ability to detect rare and common genetic variations associated with neurological disorders. These technologies, including whole-exome sequencing (WES) and whole-genome sequencing (WGS), have facilitated the identification of pathogenic mutations in genes such as APP, PSEN1, and PSEN2 in Alzheimer’s disease (AD), as well as LRRK2, SNCA, and PINK1 in Parkinson’s disease (PD) [12,13].

Genome-wide association studies (GWASs) have identified numerous risk loci linked to AD, PD, and multiple sclerosis (MS), among others [14]. For example, APOE ε4 remains a key genetic risk factor for AD, influencing amyloid-beta aggregation, tau pathology, and neuroinflammation through mechanisms involving lipid metabolism and synaptic dysfunction [15]. Variants in TREM2 have also been associated with increased AD risk, affecting microglial response and neuroimmune regulation [16]. In PD, LRRK2 mutations impact mitochondrial homeostasis and autophagy, while SNCA mutations contribute to alpha-synuclein aggregation, a hallmark of disease pathology [17]. In MS, the HLA-DRB1 allele has been strongly associated with disease susceptibility, highlighting the role of immune system dysregulation in neuroinflammatory conditions [18].

Beyond inherited genetic mutations, transcriptomic profiling has provided crucial insights into gene expression changes associated with neurological diseases [19]. RNA sequencing (RNA-seq) studies have identified altered mRNA signatures in various neurodegenerative and neuroinflammatory disorders. Differential gene expression analysis has revealed upregulation of pro-inflammatory cytokines and downregulation of neuroprotective factors in AD and MS, supporting the role of immune activation and neuroinflammation in disease progression [20]. Additionally, non-coding RNAs, including microRNAs (miRNAs) and long non-coding RNAs (lncRNAs), have emerged as critical regulators of neurodegenerative processes [21]. miRNAs such as miR-155 and miR-146a have been implicated in neuroinflammation, while lncRNAs such as BACE1-AS modulate amyloid precursor protein (APP) processing and contribute to amyloid-beta plaque formation [22].

The integration of multi-omics approaches, combining genomic, transcriptomic, and epigenomic data, has further enhanced our understanding of neurological disease mechanisms [23] Table 1. Recent studies have leveraged single-cell RNA sequencing (scRNA-seq) to uncover cell-type-specific transcriptional changes in brain tissues affected by neurodegenerative diseases. This high-resolution approach has provided new insights into the interactions between neurons, astrocytes, and microglia, revealing how genetic risk factors influence disease progression at the cellular level [24].

### 2.2. Epigenomic and Proteomic Insights

Epigenetic modifications, including DNA methylation, histone acetylation, chromatin remodeling, and non-coding RNAs, play a critical role in neurological disease progression. These modifications influence gene expression without altering the DNA sequence, often in response to environmental factors, aging, and disease pathology. Aberrant DNA methylation has been observed in genes linked to neurodegeneration, including APP, PSEN1, and TREM2 in Alzheimer’s disease (AD), as well as SNCA and LRRK2 in Parkinson’s disease (PD) [25,26]. Specifically, hypermethylation of the APP promoter has been associated with reduced gene expression, potentially contributing to amyloid-beta (Aβ) dysregulation, while hypomethylation of PSEN1 may lead to increased amyloidogenic processing [6]. In PD, aberrant methylation of SNCA influences α-synuclein aggregation, a hallmark of the disease [27].

In addition to DNA methylation, histone modifications, such as acetylation and methylation, impact chromatin accessibility and gene transcription. Histone deacetylase (HDAC) dysregulation has been linked to cognitive decline in AD and synaptic dysfunction in PD [28]. Pharmacological inhibition of HDACs has shown neuroprotective effects by enhancing memory and reducing tau pathology [29]. Similarly, chromatin remodeling enzymes, including ATP-dependent chromatin remodelers, have been implicated in regulating neuronal plasticity and neuroinflammatory responses in multiple sclerosis (MS) [30].

Beyond epigenetics, proteomic analyses have led to the discovery of cerebrospinal fluid (CSF) and blood-based biomarkers for neurodegenerative diseases. Advanced mass spectrometry and high-throughput proteomic techniques have enabled the identification of disease-specific proteins, facilitating early diagnosis and disease monitoring. Key CSF biomarkers include phosphorylated tau (p-tau), total tau (t-tau), amyloid-beta (Aβ42/Aβ40 ratio), and neurofilament light chain (NfL) [7]. Elevated levels of p-tau and t-tau in AD correlate with tau aggregation and neuronal damage, while reduced CSF Aβ42 reflects amyloid deposition in the brain [31].

In PD, proteomic studies have identified α-synuclein oligomers and DJ-1 protein levels as potential biomarkers [10]. Additionally, NfL, a marker of axonal damage, is elevated in various neurodegenerative diseases, including AD, PD, and MS [32]. Blood-based NfL measurements offer a promising non-invasive alternative for monitoring disease progression and therapeutic response [33].

The integration of epigenomic and proteomic data into multi-omics analyses is enhancing biomarker discovery and precision medicine approaches. Combining epigenetic profiles with proteomic signatures may improve diagnostic accuracy, facilitate early disease detection, and guide personalized treatment strategies [34]. Future research should focus on validating these biomarkers in large-scale longitudinal studies and exploring their potential as therapeutic targets.

### 2.3. Liquid Biopsy and Multi-Omics Integration

Liquid biopsy technologies have revolutionized the non-invasive detection of circulating biomarkers, offering new avenues for diagnosing and monitoring neurological diseases. Unlike traditional biopsies that require invasive sampling of brain tissue or cerebrospinal fluid (CSF), liquid biopsies analyze circulating molecular components in blood, plasma, saliva, or urine. These components include exosomal RNA, cell-free DNA (cfDNA), circulating tumor DNA (ctDNA), extracellular vesicles (EVs), proteins, and metabolites, providing a snapshot of disease pathophysiology in real time [3] (Figure 1). 

One of the most promising aspects of liquid biopsy in neurology is its application in neurodegenerative disorders such as Alzheimer’s disease (AD), Parkinson’s disease (PD), amyotrophic lateral sclerosis (ALS), and multiple sclerosis (MS). In AD, cell-free mitochondrial DNA (cf-mtDNA) levels have been correlated with disease progression, reflecting neuronal damage [35]. Additionally, exosomal tau and amyloid-beta peptides in blood have been proposed as alternative biomarkers for early AD diagnosis [36]. In PD, exosomal α-synuclein in plasma distinguishes PD patients from healthy individuals, serving as a potential diagnostic marker [8].

Moreover, extracellular vesicles (EVs), which carry microRNAs (miRNAs), proteins, and lipids, play a pivotal role in disease pathogenesis and biomarker discovery. For example, miR-21 and miR-155 levels in circulating exosomes are elevated in MS patients, highlighting their role in neuroinflammation [37]. Similarly, liquid biopsy analysis of circulating inflammatory cytokines and neurofilament light chain (NfL) provides valuable insights into disease severity and treatment response in neurodegenerative and neuroinflammatory conditions [38].

Beyond individual biomarkers, multi-omics integration has emerged as a powerful approach for enhancing biomarker discovery and disease classification. By combining genomic, transcriptomic, proteomic, epigenomic, and metabolomic data, researchers can identify complex interactions that underlie neurological diseases. For example, in AD, integrating GWAS data with transcriptomic and proteomic datasets has led to the identification of novel risk genes and pathways involved in neurodegeneration [33].

Multi-omics analyses have also been applied in drug development and precision medicine. Integrating metabolomics with proteomics has enabled the identification of metabolic disturbances in PD, providing potential therapeutic targets [10]. Similarly, in MS, combining DNA methylation profiles with proteomic signatures has improved our understanding of immune dysregulation and personalized treatment approaches [39].

The application of artificial intelligence (AI) and machine learning in multi-omics research has further accelerated biomarker discovery. AI-driven algorithms can process large-scale datasets to uncover hidden patterns and predict disease risk, progression, and therapeutic responses [40]. Future research should focus on standardizing liquid biopsy protocols, validating biomarkers in large-scale clinical trials, and integrating multi-omics data into clinical practice to advance precision neurology.

Additionally, regulatory approval and standardization remain critical challenges in translating biomarkers into clinical practice. While some biomarkers, such as CSF-based tau and Aβ42/Aβ40 ratio, have received regulatory validation (e.g., FDA-approved assays for Alzheimer’s disease diagnosis), others require further validation through multicenter clinical trials. The European Medicines Agency (EMA) and the U.S. Food and Drug Administration (FDA) emphasize the importance of reproducibility and large-scale cohort validation for biomarker approval [41].

Future research should prioritize:-Large-scale multi-omics integration with well-characterized patient cohorts.-Standardization of biomarker assessment protocols across different populations.-AI-driven approaches to analyze high-dimensional biomarker data, improving diagnostic accuracy.-Translation of biomarker panels into clinical decision-making tools through regulatory harmonization [5].

## 3. Discussion

The identification of molecular biomarkers has significantly improved our ability to diagnose, prognosticate, and monitor neurological diseases, allowing for earlier detection and more precise therapeutic interventions. The advent of multi-omics technologies has revolutionized the field by integrating genomic, transcriptomic, proteomic, metabolomic, and epigenomic data, leading to a more comprehensive understanding of disease mechanisms [4].

In addition to the general discussion above, a more detailed examination of the biomarkers listed in Table 1 is provided here. For example, in Alzheimer’s disease (AD), protein-based biomarkers such as phosphorylated tau (p-Tau) and the Aβ42/Aβ40 ratio are typically measured in cerebrospinal fluid (CSF) using sensitive immunoassays, with detection limits in the low pg/mL range [31]. Their diagnostic utility is closely linked to their roles in amyloid plaque formation and neurofibrillary tangle development, although pre-analytical variability and potential false positives remain challenges [7]. Similarly, neurofilament light chain (NfL) is detected using advanced techniques such as the single molecule array (Simoa) method, providing reliable quantification across AD, Parkinson’s disease (PD), multiple sclerosis (MS), and amyotrophic lateral sclerosis (ALS) [33]. In PD, biomarkers like α-synuclein and DJ-1 are measured not only in CSF but also in peripheral tissues (e.g., saliva, skin biopsies), reflecting their widespread expression; however, the overlap with other synucleinopathies can complicate interpretation [17]. For MS, the detection of CXCL13 and microRNA-21 (miR-21) via quantitative PCR and immunoassays offers insights into immune activation, yet assay standardization and the risk of false positive results need further refinement [9]. Finally, in ALS, TDP-43 and SOD1 serve as key markers of neurodegeneration, although technical challenges in reliably quantifying these proteins at very low concentrations in biological samples limit their current clinical utility [8]. Overall, while these biomarkers hold great promise for early diagnosis and prognosis, ongoing improvements in detection techniques and standardization are essential to fully realize their potential.

PET imaging offers a valuable in vivo view of the brain’s functional and metabolic changes, quantifying amyloid or tau deposition and glucose metabolism in neurodegenerative disorders despite its high cost. Meanwhile, molecular biomarkers—such as NfL, phosphorylated tau, and various genetic signatures—can detect subtle pathological changes even before they are visible on PET scans. Liquid biopsy techniques further provide a non-invasive, cost-effective monitoring alternative. In short, PET delivers essential spatial data, while molecular biomarkers promise earlier detection and systemic insights; together, they can enhance early diagnosis and guide more precise interventions.

One of the key challenges in biomarker research is ensuring clinical translation and standardization. While several promising biomarkers have been identified for Alzheimer’s disease (AD), Parkinson’s disease (PD), multiple sclerosis (MS), and amyotrophic lateral sclerosis (ALS), many remain in the experimental phase due to limited reproducibility, small sample sizes, and lack of validation in diverse populations [3]. Large-scale, longitudinal studies with standardized protocols are essential for biomarker validation [33].

Furthermore, liquid biopsy and non-invasive biomarker detection are emerging as critical components of precision neurology. Circulating biomarkers such as cell-free DNA (cfDNA), exosomal RNA, neurofilament light chain (NfL), and inflammatory cytokines have shown promise in tracking disease progression and response to treatment [8]. However, the sensitivity and specificity of these biomarkers vary, necessitating multi-omics integration for improved diagnostic accuracy [34].

Another transformative advancement is the application of artificial intelligence (AI) and machine learning (ML) algorithms in biomarker discovery and disease classification. AI-driven approaches can analyze vast amounts of multi-omics and imaging data to identify novel biomarker signatures, improve diagnostic predictions, and optimize personalized treatment strategies [32]. In AD, for example, deep learning models have been used to integrate neuroimaging and molecular biomarker data, improving early diagnosis and differentiation from other dementias [5]. In PD, AI-based analyses of blood-based biomarkers and voice patterns have enhanced early disease detection [42].

Despite these advancements, several challenges remain:Standardization and Validation: biomarkers must undergo rigorous validation across independent cohorts before being implemented in clinical practice [9].Inter-individual Variability: differences in genetics, lifestyle, and comorbidities influence biomarker expression, making it necessary to develop personalized reference ranges [9].Data Integration and Interpretation: multi-omics datasets generate large volumes of information that require sophisticated computational tools and robust statistical frameworks to extract clinically relevant insights [43].

Future research should focus on large-scale, multi-center collaborations to validate promising biomarkers, enhance AI-driven analytics, and integrate biomarker-based precision medicine approaches into routine neurological care. The ultimate goal is to develop non-invasive, cost-effective, and highly specific biomarkers that enable earlier intervention and improved patient outcomes [11,44].

## 4. Materials and Methods

This review was conducted through a comprehensive literature search using the PubMed, Scopus, and Web of Science databases. The search strategy included the following keywords:“molecular biomarkers”;“neurological diseases”;“genomics”;“proteomics”;“liquid biopsy”;“precision medicine”.

Studies published between 2010 and 2024 were considered for inclusion. Articles were selected based on their relevance, methodological rigor, and impact in the field. Priority was given to systematic reviews, meta-analyses, clinical trials, and large-scale multi-omics studies. The exclusion criteria included small-scale studies with low statistical power, studies lacking appropriate validation cohorts, and those published in non-peer-reviewed sources.

To ensure the reliability and reproducibility of the findings, the literature review focused on studies that met the following criteria:Genomic biomarkers: studies investigating single nucleotide polymorphisms (SNPs), gene mutations, and genome-wide association studies (GWASs) in neurological diseases.Transcriptomic biomarkers: RNA sequencing (RNA-seq) and differential gene expression studies.Epigenomic biomarkers: DNA methylation, histone modifications, and chromatin remodeling studies.Proteomic biomarkers: studies identifying cerebrospinal fluid (CSF) and blood-based protein biomarkers.Liquid biopsy: analysis of exosomal RNA, circulating cfDNA, and extracellular vesicles as diagnostic tools.Multi-omics approaches: integration of genomics, transcriptomics, proteomics, and metabolomics in biomarker discovery.AI and machine learning in biomarker analysis: studies utilizing AI-driven approaches for biomarker identification and clinical applications.

Data extraction and synthesis followed PRISMA (Preferred Reporting Items for Systematic Reviews and Meta-Analyses) guidelines, ensuring a transparent and reproducible selection process. Future systematic reviews may further benefit from network meta-analysis and real-world evidence from biobank datasets and electronic health records (EHRs).

## 5. Conclusions

The future of neurological disease diagnostics lies in integrating AI-enhanced multi-omics, liquid biopsy techniques, and standardized regulatory frameworks. By leveraging these advancements, precision neurology will facilitate earlier disease detection, personalized treatment strategies, and improved patient outcomes. Collaborative efforts between researchers, clinicians, and regulatory bodies are essential to bridge the gap between biomarker discovery and clinical implementation.

Molecular biomarkers are revolutionizing the landscape of neurological disease diagnostics and prognostics, providing deeper insights into disease mechanisms, early detection, and personalized therapeutic strategies. The integration of multi-omics approaches, incorporating genomics, transcriptomics, proteomics, metabolomics, and epigenomics, has enhanced our ability to identify novel disease-specific markers. This allows for greater diagnostic accuracy, improved patient stratification, and refined treatment response predictions.

One of the most transformative advancements in biomarker research is the use of multi-omics integration to uncover complex molecular interactions. Genome-wide association studies (GWASs) and transcriptomic profiling have helped elucidate the genetic basis of neurological diseases, while proteomic and metabolomic analyses have identified circulating biomarkers relevant for early disease detection. For instance, neurofilament light chain (NfL) has emerged as a robust biomarker of neurodegeneration, providing valuable insights into disease progression in Alzheimer’s disease (AD), Parkinson’s disease (PD), multiple sclerosis (MS), and amyotrophic lateral sclerosis (ALS).

In parallel, liquid biopsy techniques are reshaping non-invasive biomarker discovery, allowing for the detection of cell-free DNA (cfDNA), exosomal RNA, microRNAs (miRNAs), and extracellular vehicles (EVs) in various biofluids, including blood, cerebrospinal fluid (CSF), and plasma. These circulating biomarkers offer the advantage of real-time disease monitoring and have been successfully utilized in the early detection of AD, PD, and MS. For example, exosomal tau and amyloid-beta peptides have shown diagnostic potential for AD, while plasma α-synuclein and DJ-1 protein levels serve as promising biomarkers for PD.

Beyond individual biomarkers, the integration of artificial intelligence (AI) and machine learning (ML) algorithms has significantly enhanced biomarker discovery. AI-driven computational tools can analyze high-dimensional multi-omics datasets, enabling early disease prediction, biomarker validation, and precision medicine applications. In AD, deep learning models combining genetic, transcriptomic, and imaging data have been shown to improve diagnostic accuracy, while, in PD, AI-based voice analysis has facilitated early disease detection.

Despite these advancements, several challenges remain:Standardization and Validation: large-scale multi-center studies are needed to validate molecular biomarkers across diverse populations before clinical implementation.Inter-Individual Variability: biomarker expression varies based on genetics, environment, and comorbidities, highlighting the need for personalized reference ranges.Data Integration and Interpretation: the complexity of multi-omics datasets requires advanced bioinformatics tools and robust statistical models to extract meaningful insights.

Future research should focus on optimizing biomarker panels, refining AI-driven analytics, and establishing clinical guidelines for biomarker-based diagnostics. The convergence of multi-omics technologies, liquid biopsy innovations, and AI-driven analytics is transforming precision neurology, paving the way for earlier disease detection, improved monitoring, and more effective, individualized treatments (Table 2). 

## Figures and Tables

**Figure 1 ijms-26-02231-f001:**
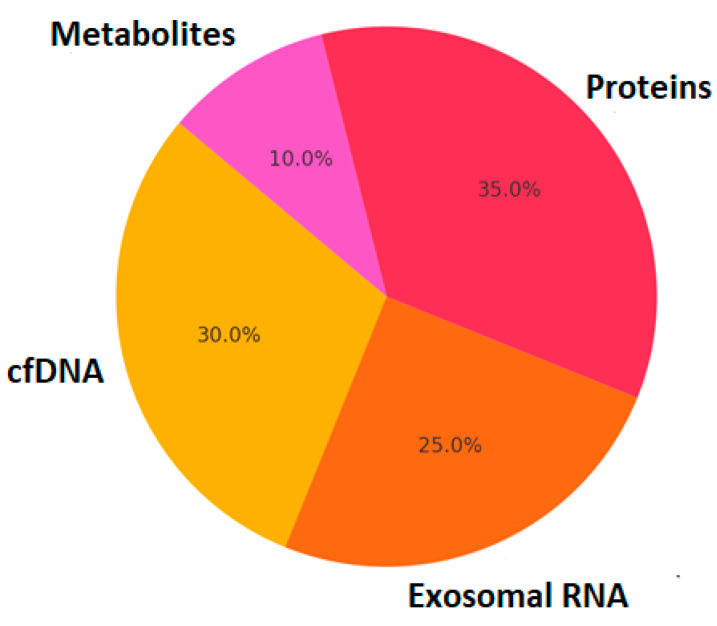
Contribution of different biomarkers in liquid biopsy.

**Table 1 ijms-26-02231-t001:** Summary of molecular biomarkers.

Disease	Biomarkers	Type	Clinical Utility
Alzheimer’s Disease (AD)	p-Tau, Aβ42/Aβ40, NfL	Protein-based, Genetic	Early diagnosis, Monitoring
Parkinson’s Disease (PD)	α-synuclein, DJ-1, NfL	Protein-based, Extracellular vesicles	Differential diagnosis, Disease progression
Multiple Sclerosis (MS)	NfL, CXCL13, miR-21	Protein-based, RNA-based	Immune profiling
Amyotrophic Lateral Sclerosis (ALS)	TDP-43, NfL, SOD1	Protein-based, Genetic	Neurodegeneration marker

**Table 2 ijms-26-02231-t002:** Technologies and platforms for biomarker analysis.

Technology	Application	Advantages
RNA-seq	Transcriptomic analysis	High sensitivity
Mass Spectometry	Proteomic analysis	Accurate protein quantification
AI-driven analysis	Predictive modeling and diagnostics	Big data analysis
Single-cell sequencing	Single-cell molecular characterization	Individual cell characterization

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
