# Peer review of "Molecular Biomarkers in Neurological Diseases: Advances in Diagnosis and Prognosis"

_ijms, 2025, doi:10.3390/ijms26052231_

Round 1
Reviewer 1 Report
Comments and Suggestions for Authors
This is a well-written overview on the state of the art of the use of molecular biomarkers to diagnose four neurological disorders (AD, MS, PD and ALS).
Minor points:
1.Could you please write a few words about PET imaging studies? Despite the fact that PET is fairly expensive - do molecular biomarkers identify disease-related pathologic changes earlier and/or more versatile than PET?
2. What about VILIP-1 in AD? I found several papers about this protein in CSF as having diagnosting value.
3. Early diagnoses of disorders are nice, but do they bring about any progress in treatment yet?
Author Response
Thank you very much for taking the time to review this manuscript. Please find the detailed responses below and the corresponding revisions/corrections highlighted/in track changes in the re-submitted file.
Comments 1: Could you please write a few words about PET imaging studies? Despite the fact that PET is fairly expensive - do molecular biomarkers identify disease-related pathologic changes earlier and/or more versatile than PET?
Response 1: Thank you for pointing this out. I agree with this comment. Therefore, I have added a few words to the Discussion in the second paragraph on page 6 in the revised manuscript attached.
Comments 2. What about VILIP-1 in AD? I found several papers about this protein in CSF as having diagnosting value.
Response 2: Visinin-like protein-1 (VILIP-1) is indeed emerging as a promising biomarker for Alzheimer’s disease (AD) in cerebrospinal fluid (CSF). Here are some key points regarding its diagnostic value:
- Indicator of Neuronal Injury: VILIP-1 is a neuronal calcium sensor protein, and elevated levels in CSF are thought to reflect neuronal injury or degeneration, which is a hallmark of AD.
- Diagnostic Utility: Several studies have reported that CSF VILIP-1 levels are significantly higher in AD patients compared to healthy controls. This elevation can complement other established biomarkers (like tau and amyloid-beta ratios) in improving diagnostic accuracy.
- Disease Progression: Some research suggests that VILIP-1 might also correlate with disease severity and could potentially predict the progression from mild cognitive impairment (MCI) to AD.
- Complementary Role: While PET imaging and other biomarkers offer spatial and functional insights, VILIP-1 provides additional biochemical evidence of neurodegeneration. In a clinical setting, using a panel that includes VILIP-1 alongside other markers may enhance overall diagnostic precision.
In summary, while VILIP-1 shows significant promise as a CSF biomarker for AD, further validation in larger, standardized studies is necessary to fully establish its role in routine clinical practice.
Comments 3. Early diagnoses of disorders are nice, but do they bring about any progress in treatment yet?
Response 3: Early diagnosis plays a crucial role in managing neurological disorders by enabling timely interventions, closer monitoring, and more informed clinical decision-making. However, its direct impact on treatment outcomes is still evolving. Here are some key points to consider:
- Timely Intervention: Early detection can allow clinicians to implement interventions sooner, potentially slowing disease progression or mitigating symptoms before more severe damage occurs.
- Current Therapeutic Landscape: For many neurodegenerative diseases, such as Alzheimer’s disease, existing treatments are largely symptomatic rather than curative. While early diagnosis can improve quality of life and provide a window for experimental therapies, it has not yet led to a breakthrough in fundamentally altering disease progression.
- Clinical Trials and Research: Early diagnosis is vital for identifying candidates for clinical trials. By diagnosing patients at an earlier stage, researchers can test disease-modifying treatments and potentially discover interventions that could change the disease course.
- Personalized Medicine: The integration of early diagnostic biomarkers with multi-omics and AI-driven approaches is paving the way toward more personalized treatment strategies. As our understanding deepens, early diagnosis may eventually translate into tailored therapies that target specific disease mechanisms.
In summary, while early diagnosis is a significant step forward and has already improved patient management and research opportunities, the translation into effective, disease-modifying treatments remains a work in progress.
Reviewer 2 Report
Comments and Suggestions for Authors
The authors in this work describe a literature survey for neurological biomarkers by identifying and categorizing different biomarkers from various studies. The authors observed that there 3 key identities to the biomarkers based on the approaches used, 1)Genomics and Transcriptomics, 2) Epigenomic and Proteomics, 3) Liquid Biopsy. I think the work is presented well and attempts to describe useful approaches for biomarker identification for hihgly critical neurological diseases and is impactful.
While the authors provide an insightful short review, the article is too short in its current form to be a review article on its own and therefore, my comments mainly focussed on further expanding it to provide more details on the biomarkers discussed by the authors. I don't think there is any scientific deficiency in the article however, I think there should be a more detailed discussion regarding the biomarkers, their detection technique, limit of detection/conc in biological samples which renders them useful in more detail, the tissues they are found in, some details on the biochemistry of the biomarkers which makes them ideal markers for the disease and prognosis. Furthermore, it would also be useful to include some potential disadvantages of these biomarkers ( false positive detection etc) which would make this article much more comprehensive and useful to a more broad scientific community. Therefore, I recommend the authors provide a paragraph or individual discussion on some or all of the biomarkers highlighted in Table 1 with the details mentioned above. This would provide more practical information while still keeping the article focussed.
Author Response
Response to Reviewer X Comments
Thank you very much for taking the time to review this manuscript. Please find the detailed responses below and the corresponding revisions/corrections highlighted/in track changes in the re-submitted file.
Comments 1: The authors in this work describe a literature survey for neurological biomarkers by identifying and categorizing different biomarkers from various studies. The authors observed that there 3 key identities to the biomarkers based on the approaches used, 1)Genomics and Transcriptomics, 2) Epigenomic and Proteomics, 3) Liquid Biopsy. I think the work is presented well and attempts to describe useful approaches for biomarker identification for hihgly critical neurological diseases and is impactful.
While the authors provide an insightful short review, the article is too short in its current form to be a review article on its own and therefore, my comments mainly focussed on further expanding it to provide more details on the biomarkers discussed by the authors. I don't think there is any scientific deficiency in the article however, I think there should be a more detailed discussion regarding the biomarkers, their detection technique, limit of detection/conc in biological samples which renders them useful in more detail, the tissues they are found in, some details on the biochemistry of the biomarkers which makes them ideal markers for the disease and prognosis. Furthermore, it would also be useful to include some potential disadvantages of these biomarkers ( false positive detection etc) which would make this article much more comprehensive and useful to a more broad scientific community. Therefore, I recommend the authors provide a paragraph or individual discussion on some or all of the biomarkers highlighted in Table 1 with the details mentioned above. This would provide more practical information while still keeping the article focussed.
Response 1: Thank you for your constructive suggestions. I have attached the revised manuscript, which now includes an additional paragraph at the end of page 6 in the Discussion that expands on the biomarkers in Table 1, detailing their detection techniques, limits of detection, tissue localization, biochemical properties, and potential drawbacks such as false positives.
Round 2
Reviewer 2 Report
Comments and Suggestions for Authors
The authors have addressed my concerns and should be accepted for publication.